# Enhancing the Production Performance and Nutrient Utilization of Laying Hens by Augmenting Energy, Phosphorous and Calcium Deficient Diets with Fungal Phytase (*Trichoderma reesei*) Supplementation

**DOI:** 10.3390/ani14030376

**Published:** 2024-01-24

**Authors:** Shoaib Ahmed Pirzado, Guohua Liu, Muhammad Adanan Purba, Huiyi Cai

**Affiliations:** 1Key Laboratory of Feed Biotechnology of Agricultural Ministry, Feed Research Institute Chinese Academy of Agricultural Sciences, Beijing 100081, China; sapirzadu@sau.edu.pk (S.A.P.); muhammadadananpurba@yahoo.co.id (M.A.P.); caihuiyi@caas.cn (H.C.); 2Department of Animal Nutrition, Sindh Agriculture University, Tandojam 70060, Pakistan

**Keywords:** phytase, laying performance, nutrient utilization, laying hen, egg quality

## Abstract

**Simple Summary:**

Supplementing laying hen diets with fungal phytase derived from *Trichoderma reesei* can be a beneficial strategy for improving production and nutrient utilization, particularly in cases where hen diets are deficient in energy, phosphorus, and calcium. The phytase enzyme enhances the birds’ ability to digest and absorb essential nutrients, particularly phosphorus and calcium, which are crucial for eggshell formation and overall health. This supplementation addresses deficiencies in conventional diets, promoting optimal egg production and quality. By revealing the nutritional potential of feed through fungal phytase, this approach not only benefits laying hens’ performance but also contributes to more sustainable and cost-effective poultry farming practices.

**Abstract:**

A ten-week trial was conducted to evaluate the enhancement of production performance and nutrient utilization of laying hens through augmenting energy, phosphorous, and calcium deficient diets with fungal phytase (*Trichoderma reesei*) supplementation. 720 Hy-line Brown hens aged 28 weeks were randomly divided into 5 groups; each group had 8 replicates of 18 hens. Five experimental diets were prepared and fed to corresponding groups. A positive control (PC) diet contained 3.50% of calcium (Ca), 0.32% of non-phytate phosphorus (NPP), and apparent metabolic energy (AME) of 11.29MJ/kg, while a negative control (NC) diet contained 3.30% of Ca, 0.12% of NPP, and lower AME of 300 kJ/kg. The other three diets were supplemented with 250 FTU/kg phytase (PHY-250), 1000 FTU/kg phytase (PHY-1000), and 2000 FTU/kg phytase (PHY-2000) in addition to a regular NC diet. Results indicated that the positive control (PC) diet group had higher body weight gain, egg weight, and average daily feed intake. However, laying rate, egg mass, and FCR were most improved in the PHY-2000 group, followed by the PHY-1000 and PHY-250 groups (*p* < 0.05). Improved yolk color was most notable in laying hens fed the diet with PHY-1000 as opposed to the PC and NC groups (*p* < 0.05), but no overall difference was found among all of the phytase treated groups. The apparent availability of dry matter, energy, phosphorus, and phytate P was significantly higher in the PHY-2000 group than in the PC and NC groups (*p* < 0.05). Compared to the PC group, nitrogen retention was significantly higher in the PHY-1000 group, while calcium availability was higher in the PHY-250 group. The results suggested that the addition of phytase to diets with low P, Ca, and AME improved laying performance and apparent availability of dietary nutrients. Thus, it was concluded that the laying hen diet could be supplemented with 1000–2000 FTU/kg phytase for improving laying production and nutrient availability and mitigating the negative impact of reduced nutrient density in laying hen diets.

## 1. Introduction

Phosphorus plays a major role in poultry feed. It is commonly present in plant ingredients in the form of phytate and is available in smaller quantities for absorption in the digestive tract of poultry due to the lack of endogenous phytase. The presence of phytate in monogastric animal feed reduces the absorption of phosphorus as well as other nutrients such as calcium, protein, energy, amino acids, trace elements, and starch because of chelation by phytate activity. Thus, phytate is usually thought to be an anti-nutritional factor for non-ruminant animals [1,2,3,4,5]. Considering the detrimental role of phytate in monogastric animals, phytase enzyme has been introduced to curtail phytate activity. This acts as a specific enzyme that hydrolyzes phytate, degrades it into an inositol hexaphosphate structure, and releases inorganic phosphorus, as well as other minerals such as calcium, manganese, and zinc, for absorption in the gut [6,7,8]. Additionally, phytase may also improve the availability of protein, amino acids [9], and energy [10] in the poultry diet. Some studies have revealed that laying hens fed a soybean and corn-based diet with phytase improved calcium and phosphorus absorption. Moreover, they also improved growth performance, laying performance, and retention of nutrients, especially minerals in blood and bones [11]. According to the reports of Jondreville et al. [12] and Lalpanmawia et al. [13] microbial phytase decreases phosphorus emission in the feces of poultry birds. Additionally, inclusion of phytase in poultry feed is beneficial for the environment, considering that dietary phytate phosphorus not utilized by animals could cause serious environmental pollution through means of fecal phosphorus [14].

Thus, phytase has special significance for the poultry industry, which is under strong pressure to reduce levels of dietary phosphorus since its excretion triggers ecological deterioration. In recent years, considering the low price of phytase, many scientists have tried adding high doses of phytase to the diets of broilers and laying hens. Cowieson et al. [15] reported that the use of high doses of phytase above 1000 U/kg of diet promoted greater nutrient availability in the diets of broiler chickens compared with diets containing lower phytase. Improved egg laying performance was recorded with addition of 20,000 FTU/kg phytase in laying hens [16]. These findings extend the application and value of phytase in animal feed.

There are many reports focusing on the effect of phytase on laying hens, but unfortunately, most of them emphasize the role of phytase in the release of phosphorus from phytate in plant feedstuffs, as well as nitrogen, calcium, and other mineral elements, and dietary deficiency of phosphorus, protein, and calcium has been used to evaluate the effect of phytase on these nutrients. However, there are scarce reports regarding the relationship between phytase and dietary energy efficiency. Li et al. [17] reported that adding phytase into low AME diets reduce them by only 60 kcal/kg. Therefore, the present study was undertaken to enhance the production performance and nutrient utilization of laying hens by augmenting energy, phosphorous, and calcium deficient diets with fungal phytase (*Trichoderma reesei*) supplementation, and to enhance understanding of the nutritional benefits of phytase in poultry feed.

## 2. Materials and Methods

### 2.1. Animals, Diets, Experimental Design, and Management

A total of 720 Hy-line Brown commercial laying hens aged 28 weeks were randomly divided into five groups, each of which had eight replicates, including 18 laying hens living in six cages. The trial started at 22 weeks old, after a one-week adaptation period, and lasted for ten weeks. The birds were kept in a 16 h photoperiod by artificial lighting. Mash feed and water were provided to the hens *ad libitum*. The environmental conditions during the trial were automatically controlled and appropriate for the hens.

The experimental diets were formulated based on corn, soybean meal, and corn by-product. A positive control diet (PC) was designed with an AME of 11.29 MJ/kg and a nutrient ratio of 3.50% Ca, 0.60% P, and 0.34% non-phytate phosphorus. A negative control diet (NC) was formulated by reducing nutrient level; this diet contained an AME of 10.99 MJ/kg, 3.25% Ca, 0.38% P, and 0.14% non-phytate phosphorus. The other three experimental diets were established through adding 250, 1000, and 2000 FTU/kg microbial phytase (PHY) to the negative control diet formulation. The formulation and nutritional levels of diets are shown in Table 1.

The sample of fungal phytase, named Axtra○R PHY G, was derived from *Trichoderma reesei* and was provided by Genencor (Wuxi) Bio-Products Co., Ltd., Jiangsu, Wuxi, China. The product contains 5000 FTU/kg phytase activity and wheat powder as a diluent.

### 2.2. Indicators Determination and Data Collection

At the beginning of the trial, all birds were weighed to obtain their initial body weight. The number of eggs produced, egg mass, and input of diet were recorded daily based on replicates. At the end of the experiment, the final body weight of the hens and remainder feed were measured, and body weight gain (BWG), average daily feed intake (ADFI), laying rate, average daily egg mass, average egg weight, and feed conversion rate (FCR) were calculated. During the last two days of the experiment, 20 eggs were randomly collected from each replicate, and egg weight, eggshell color, albumen height, egg yolk color, and eggshell thickness were measured by the QC egg gauge (TSS-YORK,, York, UK), and eggshell strength was measured by the CLB3 closed-loop hardness gauge (Instron, London, UK). In the last week of the trial, 1.4% titanium dioxide (TiO_2_) was added to all diets as an indicator to determine the availability of nutrients. Excreta was collected for three consecutive days, dried at 65 °C for 72 h, ground, and passed through a 0.4 mm sieve. Diet and excreta samples were analyzed for dry matter (DM) via oven drying (method 2001.12), nitrogen via combustion (method 990.03) AOAC [18], and gross energy via bomb calorimeter (model: C 2000, IKA, Königswinter, Germany), while calcium and phosphorus content was determined through use of an atomic absorption spectrometer (model: novAA 400 P, Analytikjena, Jena, Germany) and UV-VIS spectrophotometer (model: 1780, Shimadzu, Kyoto, Japan), respectively. The TiO_2_ content of feed and excreta was determined using the method reported by Titgemeyer et al. [19].

The availability of nutrients mentioned above was calculated using the nutrient-to-marker ratio in the diet and feces according to the following equation:Availability of nutrients (%) = [1 − (TD×TF)/(NF×ND)] × 100
TD = titanium in diet,                             TF = titanium in feces,
ND = nutrient in feces,                            ND = nutrient in diet.

### 2.3. Statistical Analysis

The obtained data was analyzed via one-way ANOVA using IBM SPSS Statistics 19.0. The means and standard error mean were presented. The significant differences among means of treatment were compared by the Tukey test. The results were considered significant at *p* < 0.05. The normality of the data was assessed using the Shapiro-Wilk test.

## 3. Results

### 3.1. Production Performance of Laying Hens

Table 2 shows the production performance of laying hens fed diets with different doses of phytase. Compared with the positive control, the negative control diet reduced the feed intake, laying rate, egg weight, and egg mass, as well as body weight gain (*p* < 0.05). The addition of 250 FTU/kg phytase in the negative control diet increased body weight gain, laying rate, egg weight, egg mass, and feed efficiency in laying hens significantly (*p* < 0.05), which rendered the results for this group similar to those of the positive control, except in terms of the ADFI, which was not influenced by 250 FTU/kg phytase addition. Following an increase in added phytase from 250 FTU/kg to 1000 FTU/kg, ADFI, laying rate, and egg mass were increased significantly (*p* < 0.05), and even better results were observed in some parameters than in the positive control group. However, the feed efficiency and egg weight were not improved in this group (*p* > 0.05). When the phytase dose was increased from 1000 FTU/kg to 2000 FTU/kg, the laying hens exhibited a higher laying rate (96.31 vs. 94.62, *p* < 0.05); however, no significant change was noted in ADFI (114.50 vs. 112.97, *p* > 0.05), egg weight (55.80 vs. 55.79, *p* > 0.05), or egg mass (53.77 vs. 52.70, *p* > 0.05).

### 3.2. Egg Quality Attributes of Laying Hens

Table 3 displays the effects of phytase on egg quality. The treatment of diets had no significant effect on thickness, strength, or color of eggshell, as well as albumen height (*p* > 0.05). Interestingly, laying hens fed the negative diet with or without microbial phytase produced a significantly deeper yolk color than the positive control (*p* < 0.05).

### 3.3. Apparent Nutrient Availability of Nutrients

As shown in Table 4, the availability of nitrogen, phosphorus, and calcium in the positive control diet was significantly higher than that in the negative control (*p* < 0.05), but the availability of DM, energy, and phytate in the positive control diet was not significantly different (*p* < 0.05) to the negative control diet. The addition of phytase increased the availability of DM, energy, nitrogen, phosphorus, phytate, and calcium significantly (*p* < 0.05). Moreover, treatment groups showed significant improvements in the results as compared to the PC group (*p* < 0.05). No significant differences in results were recorded among the different phytase doses, except that 1000 and 2000 FTU/kg phytase doses yielded a higher availability of phosphorus than the 250 FTU/kg dose (*p* < 0.05).

## 4. Discussion

Phytase is an important feed enzyme mostly used in poultry feed in order to enhance phosphorus utilization, reduce feed cost, and decrease environmental pollution caused by the excretion of phosphorus in feces [20]. Nutritionists have revealed that microbial phytase can enhance the production performance of poultry birds [21] by improving utilization of calcium and amino acids [22]. Researchers have also focused on the use of high doses (i.e., super doses) of phytase in animal feed [23,24] to obtain added benefits. In the present experiment, similar results to past reports in the literature have been observed. We observed that adding phytase supplements ranging from 250 to 1000 FTU/kg to low nutrient diets improved feed intake, laying rate, egg weight, egg mass, and FCR. There was a linear increase observed in laying rate, egg mass, and FCR alongside increase in phytase dose. Our results agreed with other reports, which found that phytase had positive effects on egg laying rate [25,26,27,28,29,30,31,32], feed intake [7,8], egg weight, egg mass [31,32], and FCR in laying hens. This indicates that phytase overcomes the negative effect of low energy diets on production performance and improves energy and nutrient efficiency.

We have revealed that in comparison to normal doses, increased doses of phytase (250, 1000, and 2000 FTU/kg) further improved laying rate, egg mass, and feed efficiency by 1.94%, 2.03%, and 0.65% respectively in the present experiment. This confirms that the super dose effect of phytase is beneficial for production in spite of its limited marginal profit. Thus, we speculated that supplementation of 2000 FTU/kg phytase in diet may improve P utilization, which likely leads to an improvement in egg laying rate and egg mass. Contrarily to this, the addition of microbial phytase showed no effect on eggshell and egg quality in our study. Our results agree with the findings of Taylor et al. [33] who stated that supplementation of phytase at levels between 300 and 1500 FTU/kg in fed diets did not affect egg quality in laying hens. Similarly, Ravindran et al. [34] found that diets comprising 250 FTU/kg of phytase and non-phytase P levels of 0.21%, 0.16%, or 0.11% have no effect on eggshell hardness (strength) and eggshell thickness. Kim et al. [16] also reported that the addition of various high levels of phytase at 10,000, 20,000, and 30,000 FTU/kg has no positive influence on egg quality. In contrast with the results mentioned above, some previous studies reported that phytase supplementation increased the egg quality with respect to eggshell hardness and thickness because of improved phosphorus and calcium availability [29,30,31,32,33,34]. These reports [32,33,34] also found that the use of phytase in 1.5 g/kg low phosphorus diets improved shell thickness and egg quality. The inconsistency of different reports in the literature could be due to the different ages of experimental hens and varying dietary levels of Ca and available P. However, no direct evidence has confirmed this hypothesis, and more studies might be necessary to clarify the effect of phytase on egg quality.

Similarly to laying performance, the improvement in the availability of nutrients facilitated by phytase has been well-reported. In the present experiment, phytase increased the availability of dry matter, energy, nitrogen, phytate, calcium, and phosphorus significantly, which agrees with numerous previous reports [7,28,32]. However, the availability did not increase persistently in accordance with the dosage of phytase. For phosphorus, the maximum of its availability was observed at 2000 FTU/kg of phytase, without significant difference compared to 1000 FTU/kg. Conversely, Cowieson et al. [15] stated that 2400 and 24,000 FTU/kg of phytase addition enhanced phosphorus availability compared to 1200 FTU/kg in a broiler experiment. For energy and nitrogen, the peak of availability was achieved at 1000 FTU/kg of phytase, while 2000 FTU/kg of phytase increased the availability of dry matter numerically. Moreover, the digestibility of DM, calcium, and phosphorus was improved by increasing to high levels of added phytase [32]. The inclusion of phytase in the diet of laying hens has been shown to improve the availability of phytate, phosphorus, and other minerals such as calcium and zinc [7], improve the retention of calcium, phosphorus, magnesium, iron, and zinc, and reduce the excretion of phosphorus in feces [28]. In laying hen diets, phytase is usually supplemented at 300 FTU/kg, although recent studies have shown that an increased phytase dose in laying hen diets can further increase phytate P degradation and ileal P digestibility and reduce P excretion [35]. Therefore, phytase degrades dietary phytate which improves the availability and digestibility of phosphorus and reduces the excretion of fecal P, which is very important in relation to environmental pollution.

Recent understanding regarding the usage of phytase in animal nutrition and its benefits has described an equivalent relationship that converts phytase dosages into nutrients, especially available phosphorus or inorganic phosphorus. According to this experiment, the addition of 1000 FTU/kg phytase to the negative control group diet led to similar egg mass results to the positive control group. This means that 1000 FTU/kg of phytase is equivalent to 300 kJ AME, as well as 2 g of inorganic phosphorus and 2.5 g calcium. It has been reported that 1000 FTU/kg of phytase supplement is equal to 2.395 g of calcium [36] and 1.66 g [37] of inorganic P. Thus, our results are in line with these research findings. Moreover, it is worth noting that feed intake was higher in the positive control group in the present study, which could prove that the phosphorus is the limiting factor in the performance of laying hens. Many reports present that the retention of calcium [7], phosphorus [34,35,36], energy [38,39,40], and amino acids [41,42] is enhanced in layers when phytase is added to the diet. The results of the present experiment agree with these research findings. Liu et al. [39] reported that the use of 300 FTU/kg phytase in laying hen diets improved the digestibility of Ca, P, N, and amino acids, and numerous other studies have determined the positive effect of microbial phytase on enhancing the availability of protein in poultry [43,44,45].

From our experiment and other reports, we can confirm the benefit of supplementation of hen diets with phytase at the usual dosage to improve laying production and minimize environmental impacts. Although there are some super dose effects of phytase on laying hens, its marginal profits are limited. It is necessary to evaluate the economy based on the price of phytase and improvements for laying production before adding phytase above doses of 1000 FTU/kg.

## 5. Conclusions

Dietary supplementation with elevated levels of phytase (1000–2000 FTU/kg) in laying birds, combined with reduced levels of calcium, phosphorus, and energy content, has been shown to enhance production performance and optimize nutrient utilization. Furthermore, this concentration has a negligible impact on the quality attributes of eggs, emphasizing its efficacy and reliability. Findings from the current study suggest that incorporating phytase into poultry diets is a sensible approach to minimize feed costs and improve productivity.

## Figures and Tables

**Table 1 animals-14-00376-t001:** Composition of Basal diet and nutritional level.

Ingredients	Positive Control	Negative Control
Corn	58.59	59.87
Soybean meal	8.00	8.00
Rape seed meal	8.60	8.00
Corn gluten meal	7.00	7.00
Corn germ meal	5.10	7.00
Mixed oil	1.57	-
Limestone	8.60	8.90
Calcium monophosphate	1.34	0.02
L-lysine hydrochloride (75%)	0.20	0.20
DL-methionine	-	0.01
NaCl	0.33	0.33
Sodium bicarbonate	0.17	0.17
Choline chloride (50%)	0.10	0.10
Premix ^1^	0.40	0.40
Calculated Nutritional Levels		
Metabolic energy, MJ/kg	11.29	10.99
Crude protein, %	16.50	16.50
Calcium, %	3.51	3.25
Phosphorus, %	0.60	0.38
Non-phytate phosphorus, %	0.34	0.14
L-lysine, %	0.75	0.75
DL-methionine, %	0.34	0.34
TSAA, %	0.65	0.65
Calculated Values of Diets		
Metabolic energy, MJ/kg	11.31	10.97
Crude protein, %	16.49	16.50
Calcium, %	3.50	3.26
Phosphorus, %	0.61	0.38

^1^ One kilogram of complete diet contains VA 7500 IU, VD3 1500 IU, VE 30 IU, VK3 1 mg, VB1 1 mg, VB2 6 mg, VB6 3 mg, folic acid 0.3 mg, niacin 40 mg, pantothenic acid 10 mg, biotin 0.1 mg, VB12 0.01 mg, Mn 60 mg, Zn 60 mg, Fe 60 mg, Cu 8 mg, Se 0.15 mg, and I 0.3 mg.

**Table 2 animals-14-00376-t002:** Influence of fungal phytase on production performance of laying hens fed ME, Ca, and P heavy and deficient diets.

Treatment	BWG (g)	Laying Rate (%)	Egg Weight (g)	ADFI (g)	Egg Mass (g)	FCR (Feed/Egg)
PC	142.52 ^a^	93.31 ^c^	56.52 ^a^	119.33 ^a^	52.74 ^ab^	2.263 ^a^
NC	133.92 ^b^	88.43 ^d^	54.33 ^b^	110.06 ^c^	48.04 ^c^	2.291 ^a^
PHY 250 FTU/kg	140.15 ^a^	92.86 ^c^	55.37 ^a^	110.47 ^c^	51.42 ^b^	2.149 ^b^
PHY 1000 FTU/kg	138.55 ^a^	94.53 ^b^	55.75 ^a^	112.95 ^b^	52.70 ^a^	2.143 ^b^
PHY 2000 FTU/kg	140.13 ^a^	96.36 ^a^	55.80 ^a^	114.50 ^b^	53.77 ^a^	2.129 ^b^
SEM	4.45	1.46	1.30	2.23	1.18	0.041
*p*-value	0.031	0.011	0.031	0.042	0.036	0.020

Note: There is a significant difference between the means with different superscripts (*p* < 0.05). BWG = body weight gain; ADFI = average daily feed intake; FCR = feed conversion ratio.

**Table 3 animals-14-00376-t003:** Influence of fungal phytase on egg quality attributes of laying hens fed ME, Ca, and P heavy and deficient diets.

Treatment	Shell Color	Eggshell Strength (N)	Shell Thickness (mm)	Albumen Height (mm)	Yolk Color
PC	23.32	54.84	0.36	3.94	5.94 ^b^
NC	22.64	52.43	0.36	3.84	7.11 ^a^
PHY 250 FTU/kg	24.02	53.16	0.36	3.99	7.33 ^a^
PHY 1000 FTU/kg	23.58	53.43	0.36	3.96	7.19 ^a^
PHY 2000 FTU/kg	23.78	52.74	0.35	3.99	7.80 ^a^
SEM	1.58	2.97	0.01	0.25	0.35
*p*-value	0.677	0.423	0.878	0.498	0.001

Note: There is a significant difference between the means with different superscripts (*p* < 0.05).

**Table 4 animals-14-00376-t004:** Influence of fungal phytase on the apparent nutrient availability of laying hens fed ME, Ca, and P heavy and deficient diets.

Treatment	DM	Energy	Nitrogen	P	Phytate P	Ca
PC	76.93 ^b^	76.07 ^b^	73.26 ^ab^	49.73 ^c^	17.43 ^b^	58.62 ^b^
NC	77.03 ^b^	75.68 ^b^	65.54 ^c^	39.54 ^d^	15.53 ^b^	53.33 ^c^
PHY 250 FTU/kg	79.51 ^a^	78.94 ^a^	74.18a ^b^	53.38 ^b^	50.87 ^a^	63.32 ^a^
PHY 1000 FTU/kg	79.83 ^a^	79.24 ^a^	75.99 ^a^	58.20 ^a^	48.74 ^a^	63.20 ^a^
PHY 2000 FTU/kg	80.35 ^a^	79.25 ^a^	74.91 ^a^	58.87 ^a^	51.52 ^a^	61.86 ^a^
SEM	6.71	5.53	6.28	4.29	5.15	3.45
*p*-value	0.009	0.018	0.001	0.002	0.001	0.003

Note: There is a significant difference between the means with different superscripts (*p* < 0.05). DM = dry matter; P = phosphorus; Ca = calcium.

## Data Availability

All compulsory data is presented in the manuscript, and we do not have any additional data.

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
