# Peer review of "Enhancing the Production Performance and Nutrient Utilization of Laying Hens by Augmenting Energy, Phosphorous and Calcium Deficient Diets with Fungal Phytase (Trichoderma reesei) Supplementation"

_animals, 2024, doi:10.3390/ani14030376_

Round 1

Reviewer 1 Report

Comments and Suggestions for Authors

All the comments and suggestions are in the attatched file. 

Author Response

Dear Reviewer,

I'd like to extend my thanks and appreciation for your kind efforts. This is regarding my Manuscript No. animals-2799643, entitled "Enhancing the production performance and nutrient utilization of laying hens by augmenting energy, phosphorous and calcium deficient diets with fungal phytase (Trichoderma reesei) supplementation". I confirm that all corrections, suggestions and queries made by the editor and reviewers were given full consideration. All corrections and suggestions were incorporated in the manuscript file and indicated in track changes. Our response to each recommendation is mentioned below.

Comments and responses

Reviewer 1

Introduction

Suggestion 1:  Line 38. degradation would not be the world. Better absorption. Phosphorus is not degraded.

Response:        Thanks for the valuable suggestion.

Suggestion 2:  Line -40. Better to change in monogastric and feed reduces the absorption.

Response:  We changed the sentence in the revised manuscript as per reviewer suggestions.

Suggestion 3:  Line 44. Phytase

Response: Thanks for the correction. It was changed according to expert suggestions and revised in the corrected version.

Suggestion 4:  Line – 74. Trichoderma reesei in italic

Response:        Revised in manuscript as per expert suggestion.

Suggestion 5:  Suggest Tukey to analyze the treatments with phytase and negative control and Dunnett test to analyze all of them with positive control.

Response:  Thanks for the nice suggestion. The description of statistical analysis is also mentioned in the materials and method section. This study is part of our previous project, which is closed now, so it is difficult to re-evaluate the data. In our opinion the statistical analysis is sufficient to present the data.

Suggestion 6:  Line -120 Correction of spelling Tuckey

Response:  Thanks for the correction and  corrected the spelling in revised manuscript.

Suggestion 7:  Line 126. Excluding feeding

Response:  The word feeding was deleted as per suggestion  

Suggestion 8:  Line 177. energy on productive performance and improved the energy and nutrient efficiency.

Response:  Thanks for the correction. Changed in revised manuscript as per expert suggestions

Suggestion 9: Line 190. why there was an improvement on yolk color?

Response: The dietary phytase is involved in breaking down the phytic acid, resulting in the release of bound minerals, particularly iron, thereby improving bioavailability and  contributing to enhancing the yolk color in laying birds. The interaction of iron with xanthophylls in the liver leads to the formation of pigments that are deposited in            the egg yolk, resulting in a more vibrant and appealing color.

Suggestion 10: Line 205. why phytase does not improve beyond 2000 FTU? I missed the lack of a real discussion in all the manuscript. Probably because all animals have a maximum absorption rate for nutrients.

Response:  There may be a saturation point, beyond which adding more phytase does not result in a significant increase in phosphorus availability. Once a certain level of phytase activity is reached, the rate of phytic acid breakdown may not increase proportionally with higher enzyme concentrations. Other factors in the diet or the digestive environment could limit the effectiveness of phytase. For example, the presence of certain minerals, anti-nutritional factors, or specific dietary components might interfere with the activity of phytase.

Suggestion 11: Conclusion. This is not true; it was similar in all treatments, except positive control.

Response: The conclusion is revised on the basis of the study outcome. I hope this version is up to the mark for acceptance.

Reviewer 2 Report

Comments and Suggestions for Authors

In general, the experiment presented in the manuscript entitled ‘Enhancing the production performance and nutrient utilization of laying hens by augmenting energy, phosphorous and calcium deficient diets with fungal phytase (Trichoderma reesei) supplementationis interesting and the presented results are valuable.

Generally, the introduction section is well done and I have no further comments here. However, I ask the authors to supplement the methodological data because they are incomplete.

The first question concerns diets. Since the authors analysed the composition of the diets (AOAC), I am asking you to include additional information about the nutritional value of the diets based on analytical protocols, to make it clear what the differences are between the calculated and obtained data. In particular, this concerns the issue of metabolic energy. Since the difference of 0.3 MJ in ME is not very high, the quality control of the prepared feeds should verify the precision with which the desired values were obtained. The data in the feed programs for formulating diets are for information purposes only. Therefore, when creating diet recipes, you should rely on analytical data. Therefore, please add these details to Table 1.

Please indicate how you estimated the energy value of feed and excrement (e.g., regression equations, etc.)? Have you determined the gross energy in both materials? If so, please specify whether you have used a regression equation or a calorimetrical protocol (+AOAC procedure number, if appropriate)? Also, specify the nutritional recommendations according to which you formulate your diets.

L118-122: Provide information whether the numerical data were checked for normal distribution. Provide the test by which the normality of the distribution was assessed.

I have no comments to discuss the results, discussions, or conclusions. These sections are written quite solidly. However, check the text for editorial errors because there are many of them. For instance: L44 (two sentences separated by a comma), L82 (‘adlibitum’ -> ‘ad libitum’), give ‘Trichoderma reesei’ in italics, and remove all editorial errors involving the combination of numbers and units (eg, L85, L87, L135, L184, L196, L206, L226, L227), double spaces, etc.

Author Response

Dear Reviewer,

I'd like to extend my thanks and appreciation for your kind efforts. This is regarding my Manuscript No. animals-2799643, entitled "Enhancing the production performance and nutrient utilization of laying hens by augmenting energy, phosphorous and calcium deficient diets with fungal phytase (Trichoderma reesei) supplementation". I confirm that all corrections, suggestions and queries made by the editor and reviewers were given full consideration. All corrections and suggestions were incorporated in the manuscript file and indicated in track changes. Our response to each recommendation is mentioned below.

Comments and responses

Reviewer: 2

Suggestion 1: Generally, the introduction section is well done, and I have no further comments here. However, I ask the authors to supplement the methodological data because they are incomplete.

Response:  Thanks for your appreciation for the introduction section. We revised the methodological data as per expert recommendation.

Suggestion 2:  The first question concerns diets. Since the authors analysed the composition of the diets (AOAC), I am asking you to include additional information about the nutritional value of the diets based on analytical protocols, to make it clear what the differences are between the calculated and obtained data. In particular, this concerns the issue of metabolic energy. Since the difference of 0.3 MJ in ME is not very high, the quality control of the prepared feeds should verify the precision with which the desired values were obtained. The data in the feed programs for formulating diets are for information purposes only. Therefore, when creating diet recipes, you should rely on analytical data. Therefore, please add these details to Table 1.

Response: Thanks for nice suggestions. The data was analyzed according to the method of AOAC (2000) and all analytical protocols are given in the corrected version of manuscript. Moreover, 0.3 MJ is quite high and equal to (75 kcal/kg). More studies are in line to reduce the more energy up to 0.7 to 1 MJ (150 to 250 kcal/kg). We added in revised paper, as shown in Table 1.

Suggestion 3:  Please indicate how you estimated the energy value of feed and excrement (e.g., regression equations, etc.)? Have you determined the gross energy in both materials? If so, please specify whether you have used a regression equation or a calorimetrical protocol (+AOAC procedure number, if appropriate)? Also, specify the nutritional recommendations according to which you formulate your diets.

Response: Thanks for query.  We estimate the energy by bomb calorimeter of feed and excreta, and we formulate the diets according to the recommendation of NRC 1994. 

Suggestion 4: L118-122: Provide information whether the numerical data were checked for normal distribution. Provide the test by which the normality of the distribution was assessed.

Response: Information on the normality of the distribution was provided in revised version of the paper.

Suggestion 5:  I have no comments to discuss the results, discussions, or conclusions. These sections are written quite solidly. However, check the text for editorial errors because there are many of them. For instance: L44 (two sentences separated by a comma), L82 (‘adlibitum’ -> ‘ad libitum’), give ‘Trichoderma reesei’ in italics, and remove all editorial errors involving the combination of numbers and units (eg, L85, L87, L135, L184, L196, L206, L226, L227), double spaces, etc.

Response: All corrections were incorporated in revised manuscript as per suggestions. Thanks a lot for your supportive comments and valuable suggestions. The authors of the manuscript are grateful to the subject experts for valuable suggestions to improve the quality of this manuscript. We do hope that this version of the manuscript is suitable for acceptance and publication.

Reviewer 3 Report

Comments and Suggestions for Authors

Enhancing the production and nutrient utilization of laying hens by augmenting energy, phosphorus and calcium deficient with fungal phytase (Trichoderma reesei) supplementation

Dear Authors,

The manuscript is interesting and well prepared. Experiment shows that application of fungal phytase can release calcium, phosphorus and energy in dephytinization process, what decrease costs of production and contamination of environment, especially by phosphorus. There are not much to correct in the text of manuscript.

Below I add some suggestions helpful in this process:

Line 1

First capital letter C.

Line 4 and 15

Trichoderma reesei – binominal name, italics must be used.

Line 19-163

In text of manuscript is: energy (AME) 11.29MJ/kg, space after 11.29 MJ/kg needed.

Line 24-163

In text is P<0.05, must be p<0.05.

Line 47

In manuscript is [6,7,8], must be [6-8].

Line 52-163

In text is Jonderville et al [12], al. is abbreviation in this case dot must be added.

Line 74

Same like in line 4

Line 82

In text of manuscript is ‘adlibitum’, must be ‘ad libitum’.

Line 91

Table 1

Components:

In text is lysine hydrochloride, must be L-lysine hydrochloride (75%)

In text is Methionine (88%), more precise name must be used: DL-methionine or methionine hydroxy analogue?

In case of chemical composition: L-lysine and DL-methionine must be emphasized.

Line 120

Space not needed on the beginning of sequence.

Line 121

p-value must be used, for sample not for population, and so on to the end of the Results.

Line 135

Space before vs in line needed.

Line 148

Table 3.

Capital letters needed in case of first word in heading of table (Shell and Yolk)

Line 184

Space needed after 2000 and before FTU/kg phytase

Line 196

Space

Line 206

Space needed after 1000 and before FTU/kg phytase

Line 215 and 216

Without space

Line 262

Two dots on the end of sentence.

Line 263

References

In case volume italics are not required.

Line 274-354

Abbreviations needed to check from reference no. 8, 10, 11, ...

Dots in abbreviations needed [7, 9, …]

Line 355

Journal name disappeared. Volume and pages are also required.

Author Response

Dear Reviewer,

I'd like to extend my thanks and appreciation for your kind efforts. This is regarding my Manuscript No. animals-2799643, entitled "Enhancing the production performance and nutrient utilization of laying hens by augmenting energy, phosphorous and calcium deficient diets with fungal phytase (Trichoderma reesei) supplementation". I confirm that all corrections, suggestions and queries made by the editor and reviewers were all given full consideration. All corrections and suggestions were incorporated in the manuscript file and indicated in track changes. Our response against each recommendation is mentioned below.

 Comments and responses

Reviewer: 3

Suggestion 1:  Line 1 First capital letter C.

Response: Thanks for indication, I changed the small letter c into capital C for expert suggestion.

Suggestion 2:  Line 4 and 15 Trichoderma reesei – binominal name, italics must be used.

Response: Thanks for suggestions, we changed Trichoderma reesei in italic in revised manuscript.   

Suggestion 3:  Line 19-163 In text of manuscript is: energy (AME) 11.29MJ/kg, space after 11.29 MJ/kg needed.

Response: Revised as per expert suggestion.             

Suggestion 4:  Line 24-163. In text is P<0.05, must be p<0.05.

Response: Thanks for nice advice. All the correction were mentioned in revised manuscript.

Suggestion 5:  Line 47 In manuscript is [6,7,8], must be [6-8].

Response: Thanks for corrections. We changed these reference style in revised manuscript.

Suggestion 6:  Line 52-163. In text is Jonderville et al [12], al. is abbreviation in this case dot must be added.

Response:  All corrections were made in revised manuscript.

Suggestion 7:  Line 74. Same like in line 4

Response: Correct same as Line 4.

Suggestion 8:  Line 82. In text of manuscript is ‘adlibitum’, must be ‘ad libitum’.

Response:  Changed as per expert recommendation.

Suggestion 9:  Line 91 Table 1. Components: In text is lysine hydrochloride, must be L-lysine hydrochloride (75%) In text is Methionine (88%), more precise name must be used: DL-methionine or methionine hydroxy analogue? In case of chemical composition: L-lysine and DL-methionine must be emphasized.

Response: Revised in manuscript as per expert suggestions.

Suggestion 10: Line 120. Space not needed on the beginning of sequence.

Response: Thanks for nice suggestion. Space removed as per reviewer suggestions.

Suggestion 11: Line 121. p-value must be used, for sample not for population, and so on to the end of the results.

Response: Thanks for correction. Revised as per reviewer recommendation.

Suggestion 12: Line 135. Space before vs in line needed.

Response:  As per reviewer suggestion, space given before vs in revised manuscript.

Suggestion 13: Line 148. Table 3. Capital letters needed in case of first word in heading of table (Shell and Yolk).

Response:  Thanks for correction, the 1st letter of shell and yolk changed into capital letters accordingly.

Suggestion 14: Line 184. Space needed after 2000 and before FTU/kg phytase.

Response: Space given in the manuscript as per expert recommendations.

Suggestion 15: Line 196. Space

Response:  Space given in revised manuscript.

Suggestion 16: Line 206. Space needed after 1000 and before FTU/kg phytase

Response:  Space given in manuscript as per expert recommendations.

Suggestion 17: Line 215 and 216. Without space

Response:  Correct in manuscript as per reviewer suggestion.

Suggestion 18: Line 262. Two dots on the end of sentence.

Response:  Remove one dot in manuscript.

References

Suggestion 19: Line 263. References. In case volume italics are not required.

Response:  Thanks for suggestion. We fix all errors in references.

Suggestion 20: Line 274-354. Abbreviations needed to check from reference no. 8, 10, 11, ...

Dots in abbreviations needed [7, 9, …].

Response: Thanks for correction. The abbreviations are given, and dots also added in corrected version of manuscript as per expert suggestions.

Suggestion 21: Line 355. Journal name disappeared. Volume and pages are also required.

Response:  Thanks for your suggestions. We added volume and pages in references in corrected version of manuscript.

With kind regards.

Prof. Dr. Liu Guohua

E-mail:             liugouhua@caas.cn; sapirzadu@sau.edu.pk
